# Identification and Characterization of *Rhipicephalus microplus* ATAQ Homolog from *Haemaphysalis longicornis* Ticks and Its Immunogenic Potential as an Anti-Tick Vaccine Candidate Molecule

**DOI:** 10.3390/microorganisms11040822

**Published:** 2023-03-23

**Authors:** Paul Franck Adjou Moumouni, Souichirou Naomasa, Bumduuren Tuvshintulga, Nariko Sato, Kiyoshi Okado, Weiqing Zheng, Seung-Hun Lee, Juan Mosqueda, Hiroshi Suzuki, Xuenan Xuan, Rika Umemiya-Shirafuji

**Affiliations:** 1National Research Center for Protozoan Diseases, Obihiro University of Agriculture and Veterinary Medicine, Obihiro 080-8555, Hokkaido, Japan; 2Immunology and Vaccines Laboratory, C. A. Facultad de Ciencias Naturales, Universidad Autónoma de Querétaro, Carretera a Chichimequillas, Queretaro 76140, Mexico

**Keywords:** BmATAQ homologue, EGF-like domain, *Haemaphysalis longicornis*, anti-tick vaccine

## Abstract

Although vaccines are one of the environmentally friendly means to prevent the spread of ticks, there is currently no commercial vaccine effective against *Haemaphysalis longicornis* ticks. In this study, we identified, characterized, localized, and evaluated the expression patterns, and tested the immunogenic potential of a homologue of *Rhipicephalus microplus* ATAQ in *H. longicornis* (HlATAQ). HlATAQ was identified as a 654 amino acid-long protein present throughout the midgut and in Malpighian tubule cells and containing six full and one partial EGF-like domains. *HlATAQ* was genetically distant (homology < 50%) from previously reported ATAQ proteins and was expressed throughout tick life stages. Its expression steadily increased (*p* < 0.001) during feeding, reached a peak, and then decreased slightly with engorgement. Silencing of *HlATAQ* did not result in a phenotype that was significantly different from the control ticks. However, *H. longicornis* female ticks fed on a rabbit immunized with recombinant HlATAQ showed significantly longer blood-feeding periods, higher body weight at engorgement, higher egg mass, and longer pre-oviposition and egg hatching periods than control ticks. These findings indicate that the ATAQ protein plays a role in the blood-feeding-related physiological processes in the midgut and Malpighian tubules and antibodies directed against it may affect these tissues and disrupt tick engorgement and oviposition.

## 1. Introduction

*Haemaphysalis longicornis* Neumann, 1901 (Acari: Ixodidae), commonly called the Asian longhorned tick, is a three-host tick of the Metastriata family. It exhibits parthenogenetic and bisexual phenotypes and feeds on a wide variety of hosts including wildlife, livestock, companion animals, and humans. *Haemaphysalis longicornis* populations are endemic in Australia, New Zealand, New Caledonia, Fiji, the Korean Peninsula, Northeastern China, Northeastern Russia, and Japan [1,2]. Parthenogenetic phenotypes were first reported in the USA in 2017 and are now established in several states [3]. In all these regions, the Asian longhorned tick is perceived as an economic, veterinary, and public health concern because of its ability to infest hosts in large numbers, cause damage to hides, and transmit several infectious disease agents to animals and humans [1,4,5,6]. 

Methods for controlling tick populations include visual inspection of hosts and tick removal, tick habitat modifications, acaricide treatment of hosts, vegetation, equipment, and anti-tick vaccines [7]. Acaricide treatment is currently the most widely used method and several commercialized chemicals were reported effective at reducing the number of *H. longicornis* in the field or on animals [8,9,10,11]. Nevertheless, acaricides alone cannot satisfactorily reduce *H. longicornis* populations or provide comprehensive protection to livestock because the tick has a wide host range, overlapping activity periods of stadia, and spends the greater part of its annual cycle on pasture [1]. In addition, acaricides have negative effects on the environment, are expensive, and are toxic to beneficial insects. Their inappropriate usage has led to contamination of the environment and food and the occurrence of acaricide-resistant tick populations including *H. longicornis* specimens [12,13]. In contrast to acaricides, anti-tick vaccines are environment friendly, sustainable, and are suggested as the cheapest and most effective approach for the prevention and control of tick infestation and tick-borne pathogens [12]. They can be used either alone or as a component of an integrated pest management strategy [14]. Unfortunately, currently, there are no commercial anti-tick vaccines that are effective against *H. longicornis*.

A major difficulty in the development of anti-tick vaccines is the identification of tick antigens inducing an efficient protective immune response in the host. Among the numerous tick antigens tested for the development of vaccines, only Bm86, a protein expressed in the midgut of *Rhipicephalus* (*Boophilus*) *microplus* could be transformed into effective vaccines, namely TickGARD Plus™ (Intervet Australia Pty. Ltd.; Victoria, Australia; discontinued) and Gavac™ (Heber Biotec S.A.; Havana, Cuba, still in use in Latin America) [14,15]. Bm86 homologues have been characterized in several tick species including *H. longicornis*. However, the evaluation of Bm86-based vaccines showed that their efficacy varies depending on tick species and geographical location [16,17]. Current efforts to develop a new generation of anti-tick vaccines focus on identifying candidate antigens and combinations of antigens that will be more consistently effective than Bm86 [18]. Among the potential vaccine targets, ATAQ, a protein structurally related to Bm86, is an attractive candidate protective antigen.

ATAQ is a concealed protein with multiple epidermal growth factor (EGF)-like domains and a characteristic “YFNATAQRCYH” signature peptide. The function of ATAQ is unknown but it is expressed in the midgut and Malpighian tubules of all tick life stages and showed more continuous expression over stages than observed with Bm86 [16]. It has been isolated, characterized, and sequenced from several veterinary important tick species, namely *R. annulatus*, *R. decoloratus*, *R. (B) microplus*, *R. evertsi evertsi*, *R. appendiculatus*, *Amblyomma variegatum*, *H. elliptica*, *Hyalomma marginatum*, *Dermacentor reticulatus,* and *D. variabilis*. A genetic diversity analysis showed that the ATAQ proteins of the Rhipicephalinae are more conserved than this group’s Bm86 orthologues [16]. A 93.3% nucleotide identity was recorded among several species of the genus *Rhipicephalus* [19] while among *R.* (*B.*) *microplus* isolates from different regions in Mexico, the sequence identity at nucleotides and protein levels ranged between 98 and 100% and 97.8–100%, respectively [20]. Vaccine efficacy studies using the synthetic peptides showed that ATAQ peptides were moderately effective in controlling the tick life-cycle parameters. Efficiencies of 35% and 47% reduction in overall tick life cycle parameters were reported for *R.* (*B.*) *microplus* and *R. sanguineus,* respectively [19]. Although these results support the value of ATAQ as an anti-Rhipicephalinae vaccine candidate, there are no data on ATAQ in *H. longicornis* ticks. Therefore, in this study, a homologue of BmATAQ, designated as *HlATAQ,* was isolated from parthenogenetic *H. longicornis* and characterized. The transcriptional profiles in different developmental stages, organs, and feeding phases were examined, and immunostaining and gene silencing by RNA interference (RNAi) were performed to contribute to elucidating the localization and the function of HlATAQ. Recombinant proteins were expressed in bacteria, purified, and used in immunization trials to assess the potential of HlATAQ as an anti-tick vaccine candidate. Taken together, the results of this study indicate that HlATAQ plays a role in the blood-feeding-related physiological processes in the midgut and Malpighian tubules, and antibodies directed against it could disrupt tick engorgement and oviposition.

## 2. Materials and Methods

### 2.1. Ticks and Rabbits

#### 2.1.1. Ticks and Tissues Collection

Parthenogenetic *H. longicornis* ticks (Okayama strain) were used in the present study. These ticks have been maintained at the National Research Center for Protozoan Diseases, Obihiro University of Agriculture and Veterinary Medicine, for successive generations by feeding on the ears of Japanese white rabbits (Japan SLC, Shizuoka, Japan) using the cotton bags method [21]. After blood feeding, female ticks were reared under constant dark conditions at 25 °C and 100% relative humidity and allowed to lay eggs. Each consecutive life stage was also allowed to feed on rabbits. 

Whole tick specimens representing all tick life stages and tissues obtained by dissecting female ticks under the microscope were collected. Sample materials were stored at −80 °C until use.

#### 2.1.2. Rabbits

Female Japanese white rabbits (specific-pathogen-free animals, 18 weeks old purchased from Japan SLC) were used for the animal experiments. The rabbits were kept in a room with a temperature of 25 °C, humidity of 40%, and controlled lighting (period of light from 6:00 to 19:00 h). Throughout the course of the experiments, the rabbits had ad libitum access to water and commercial pellets (CR-3; CLEA Japan, Tokyo, Japan).

#### 2.1.3. Ethical Statement

The experimental design and management of animals were approved by the Experimental Animal Committee of Obihiro University of Agriculture and Veterinary Medicine (Animal experiment approval numbers: 19-74, 19-224, 20-85, 21-40, 21-226).

### 2.2. Identification of HlATAQ cDNA

An Expressed Sequence Tags (EST) database which was previously constructed in our laboratory using the midgut cDNA library of semi-engorged female *H. longicornis* [22,23,24] was searched to identify EST sharing similarities with *R. microplus* Bm86. Sequences of identified EST were examined using the BLASTX sequence homology search of NCBI (National Center for Biotechnology Information, National Institute of Health, http://blast.ncbi.nlm.nih.gov/Blast.cgi, accessed on 20 June 2019). Upon identification of the EST sharing homology with the *R. microplus* ATAQ gene, the *Escherichia* coli clone containing the corresponding plasmid DNA was selected for further analysis. 

The selected *E. coli* clone was cultured overnight in a Luria-Bertani (LB) medium with 50 mg/mL ampicillin sodium at 37 °C and its pGCAP1 plasmids were extracted using the QIAGEN Plasmid mini kit (Qiagen, Hilden, Germany). The length of the target gene was first estimated through enzymatic digestion and then submitted to sequencing. In detail, the extracted plasmids were digested with *Eco*RI and *Not*I restriction enzymes (Nippon gene, Tokyo, Japan), and the product was electrophoresed on 1.5% agarose gel, stained in ethidium bromide solution (Nacalai Tesque, Kyoto, Japan), and visualized under UV transilluminator (Printgraph AE-6905CF; Atto, Tokyo, Japan).

The sequence of the cDNA was determined by repeatedly sequencing the plasmids with pGCAP1 vector-specific primers and target gene-specific primers (Table 1).

Afterward, the obtained overlapping partial sequences were aligned using GENETYX^®^ ver. 7 (GENETYX, Tokyo, Japan), and the full-length cDNA sequence was identified. Gene-specific primers were designed using the primer walking method. All sequences were obtained by performing Sanger sequencing using the BigDye™ Terminator v3.1 Cycle Sequencing Kit (Applied Biosystems, Foster City, CA, USA) and ABI Prism 3100 Genetic Analyzer (Applied Biosystems).

To annotate the different sections of the full-length cDNA, the translate tool of the Expasy proteonomic server (http://www.expasy.ch/tools/pi_tool.html, accessed on 28 August 2019) was used to identify the Open Reading Frame (ORF), and the other sections were identified by referring to the reported features of cDNAs produced by the vector caping method [22,23]. The cDNA sequence was designated as the *H. longicornis* ATAQ gene (*HlATAQ*) and was registered in the NCBI GenBank database under the accession number: ON210133.

### 2.3. Gene Sequence and Phylogenetic Analyses of HlATAQ

The Open Reading Frame (ORF) sequence of HlATAQ was analyzed with BioEdit software (version 7.2) and translated into the amino acid sequence. The predicted molecular weight and the isoelectric point (pI) were determined using the Compute pI/MW tool of the Expasy proteomics server (https://www.expasy.org/resources/compute-pi-mw, accessed on 1 March 2023). Domain searches were performed on the amino acid sequence using PRATT version 2.1 (https://web.expasy.org/pratt/, accessed on 1 March 2023) and validated with InterPro (https://www.ebi.ac.uk/interpro/search/sequence/, accessed on 1 March 2023). SignalP 6.0 (https://services.healthtech.dtu.dk/services/SignalP-6.0/, accessed on 1 March 2023) was used for the signal peptide search. N-linked glycosylation and O-linked glycosylation of the putative protein sequences were identified using NetNGlyc 1.0 (https://services.healthtech.dtu.dk/services/NetNGlyc-1.0/, accessed on 1 March 2023) and NetOGlyc 4.0 (https://services.healthtech.dtu.dk/services/NetOGlyc-4.0/, accessed on 1 March 2023). Meanwhile, the potential glycosyl-phosphatidyl inositol (GPI) anchor sites and transmembrane (TM) helices were predicted using PredGPI (http://gpcr2.biocomp.unibo.it/predgpi/, accessed on 1 March 2023) and TMHMM-2.0 (https://services.healthtech.dtu.dk/services/TMHMM-2.0/, accessed on 1 March 2023), respectively.

The homology of HlATAQ to previously published sequences was assessed using the BLASTp algorithm of the NCBI GenBank database. All ATAQ sequences deposited in GenBank were aligned with HlATAQ and an identity/similarity matrix of the ATAQ protein family was generated using the SIAS tool (http://imed.med.ucm. es/Tools/sias.html, accessed on 1 March 2023).

A phylogenetic tree was constructed using the HlATAQ sequence, the ATAQ sequences from other tick genera, and the reference sequence of the Bm86 homologue from *H. longicornis* (Hl86) deposited in the GenBank database. Sequence alignments were created and tested with the web-based program GUIDANCE 2 [25] and the phylogenetic tree was inferred by the maximum likelihood method using MEGA X [26].

### 2.4. Analysis of HlATAQ Expression by Real-Time PCR

The expression patterns of the *HlATAQ* gene were investigated by performing real-time PCR analysis on total RNA extracted from different tick developmental stages and from adult tick midguts and Malpigian tubules. In detail, the transcription levels of *HlATAQ* were examined in eggs, unfed larvae, engorged larvae, unfed nymphs, engorged nymphs, unfed females, females at the slow feeding stage, females at the rapid feeding stage, and engorged females. To assess the expression levels in the midgut and Malpighian tubules, the tissues of female ticks at different blood-feeding phases were collected. The midgut samples were collected on day 0 (unfed), day 2 (slow feeding stage), day 4 (rapid feeding stage), and at the engorgement stage. Meanwhile, Malpighian tubules were collected on day 0, day 2, and day 6, and after detachment of fully fed ticks.

Total RNA was extracted from whole ticks and tick tissues according to the standard protocol of TRI reagent^®^ (Sigma-Aldrich, St. Louis, MO, USA). Total RNA samples were then subjected to DNase treatment using TURBO DNA-free™ (Thermo Fisher Scientific, Waltham, MA, USA), and concentrations and purity were determined with a Nanodrop^TM^ 2000 Spectrophotometer (Thermo Fisher Scientific). Thereafter, cDNA was synthesized from DNA-free RNA using the ReverTra Ace^®^ qPCR RT Kit (Toyobo, Osaka, Japan) according to the manufacturer’s directions. The cDNA was stored at −20 °C until use in real-time PCR.

The real-time PCR assays were performed according to the standard protocol using a 7300 Real-Time PCR System (Applied Biosystems) and THUNDERBIRD^®^ SYBR^®^ qPCR Mix (Toyobo). The gene-specific primer sets employed are indicated in Table 1. To calculate the relative expression levels of *HlATAQ*, standard curves were generated using two-fold serial dilutions of the cDNA of unfed female *H. longicornis* and cycling conditions comprising a 10 min heat denaturation and polymerase activation step at 95 °C followed by 40 cycles of a denaturation step at 95 °C for 15 s, and an annealing/extension step at 60 °C for 60 s. The specificity of PCR primers was confirmed by a melting curve analysis. Data were collected with the 7300 system SDS software version 1.4 for Windows (Applied Biosystems) and analyzed using Microsoft Excel [27]. The *Haemaphysalis longicornis actin*, glyceraldehyde-3-phosphate dehydrogenase gene (*GAPDH*), *L23*, and *P0* were evaluated as candidate internal control genes. As a result of evaluating the expression stability of the internal control candidate genes in all samples, *P0* was the most stable. The expression data were normalized using the *H. longicornis* ribosomal protein P0 (*HlP0*) (accession number EU048401) as the reference gene.

### 2.5. RNA Interference

RNA interference (RNAi) experiments were conducted to assess the effect of silencing *HlATAQ* on *H. longicornis* life cycle parameters. Double-stranded RNA (dsRNA) for RNAi was synthesized for two different regions of the gene. Briefly, the pGCAP1 plasmids containing the *HlATAQ* full-length cDNA were transformed into ECOS^TM^ Competent *E. coli* DH5α (Nippon Gene) which were cultured overnight. Afterward, the plasmids were extracted and purified from cultured competent cells, following the protocol of NucleoSpin^®^ Plasmid EasyPure (Takara Bio, Shiga, Japan). Oligonucleotide primers including T7 promoter sequences at the 5′end were then used to PCR- amplify two regions of *HlATAQ* (*HlATAQ-*2 (574 bp) and *HlATAQ*-4 (567 bp)) from the cDNA plasmids (Table 1). The amplifications were performed in 50-μL PCR reaction mixtures containing 5.0 μL of 10× PCR Buffer for KOD-Plus-Neo, 3.0 μL of 25 mM MgSO_4_, 5.0 μL of 2 mM dNTPs, 1.5 μL each of 10 μM forward and reverse primers, 1.0 μL of KOD-Plus-Neo polymerase (1.0 U/μL) (Toyobo), and 0.2 μL of plasmid and sterile water. PCR conditions were set at 94 °C for 2 min, followed by 40 cycles of 98 °C for 10 s, 64 °C for 30 s, and 68 °C for 15 s.

The PCR products were subjected to gel electrophoresis, then extracted and purified using the NucleoSpin^®^ Gel and PCR Clean-up kit (Takara Bio), phenol/chloroform/isoamyl alcohol (25:24:1), and 3M sodium acetate and ethachinmate (Nippon Gene). The purified DNAs were used to synthesize two dsRNAs named *HlATAQ-2* dsRNA and *HlATAQ-4* dsRNA, with a T7 RiboMax™ Express RNAi System (Promega, Madison, WI, USA) according to the standard protocol. The dsRNA of the firefly *luciferase* (*Luc*) gene [28] was used as the negative control. The size and quality of dsRNAs were confirmed by electrophoresis on 1.5% agarose gel. The dsRNA aliquots were stored at −80 °C until use.

*HlATAQ*-2 dsRNA, *HlATAQ*-4 dsRNA, or *Luc* dsRNA (1 μg/tick) was injected from the fourth coxae into the hemocoel of unfed female *H. longicornis* fixed on a glass slide with adhesive tape [29]. The injections were performed with 10 μL microcapillaries (Drummond Scientific Company, Broomall, PA, USA) drawn to fine-point needles by heating. After dsRNA injection, the ticks were left for 24 h in an incubator set at 25 °C and then simultaneously fed on the ears of Japanese white rabbits. The rabbits were monitored daily. The ticks were left to feed until engorgement and collected when they dropped from the host. Engorged ticks were put in an incubator set at 25 °C and allowed to lay eggs which were collected, transferred to individual containers on the 20th day after the start of oviposition, and incubated until hatching.

For each of the injected ticks, the length of the blood feeding period (days), pre-oviposition period (days), the oviposition to egg hatching period (Egg hatching period; days), body weight at engorgement (mg), and egg mass at 20 days after oviposition (mg) were recorded. In addition, to evaluate the efficiency of *HlATAQ* knockdown, the expression level of the *HlATAQ* gene was examined by real-time PCR using total RNAs extracted from injected ticks collected on the 4th day of blood. The real-time PCR assays were carried out as described above.

### 2.6. Localization of HlATAQ Protein by Immunohistochemistry

HlATAQ localization was performed on midgut and Malpighian tubules removed from 5-day-fed female ticks dissected under a stereomicroscope (SZX16; Olympus, Tokyo, Japan). The collected midgut and Malpighian tubules were immersed and fixed in 4% paraformaldehyde at 4 °C overnight. The next day, the tissues were immersed in PBS and left to stand at 4 °C for 24 h. Then, to prevent the formation of ice crystals during freezing, the sample mixture was immersed in a 5% sucrose solution, 10% sucrose solution, 15% sucrose solution, and 20% sucrose solution every other day and allowed to stand at 4 °C for 24 h. The samples were embedded in the OCT compound (Sakura Finetech Japan, Tokyo, Japan). Frozen sections (10 μm thick) were prepared with a cryostat (CM3050 S; Leica, Wetzlar, Germany).

The sections were air-dried for 1 h, washed with PBS, blocked with 5% skim milk (Wako, Osaka, Japan) (1 h at room temperature), and incubated with anti-HlATAQ peptide antibodies (Eurofin Genomics, Tokyo, Japan) as a primary antibody or serum of naïve mouse as control, both diluted 1: 100 with 5% skim milk solution. Alexa Fluor^®^ 594 goat anti-mouse IgG (Thermo Fisher Scientific) diluted 1: 1,000 was used as the secondary antibody. The sections were then mounted in ProLong^®^ Diamond Antifade Mountant with DAPI (Thermo Fisher Scientific), covered with a cover glass, left overnight in a dark place, and later observed under a fluorescence microscope (BZ-9000; KEYENCE, Osaka, Japan).

### 2.7. Expression and Purification of Recombinant HlATAQ

Two HlATAQ recombinant proteins, one covering the whole ORF (rHlATAQ) and the second covering a truncated portion of the ORF (rtHlATAQ), were produced in this study. To obtain rHlATAQ, *HlATAQ* was PCR-amplified using a set of forward and reverse primers (rHlATAQF, rHlATAQR; Table 1) to add restriction enzyme sites. After double digestion with *Xho*I and *Bam*HI restriction enzymes, the PCR amplicon was inserted in a pET28a plasmid (Novagen, Madison, WI, USA). After confirmation of correct insertion of HlATAQ by sequencing analysis (primers set: T7 promoter, rHlATAQ middle, T7 terminator; Table 1), the constructed plasmid was transformed into the *E. coli* BL21 (DE3) pLySs strain (Thermo Fisher Scientific) for protein expression.

To produce rtHlATAQ, the full amino acid sequence of HlATAQ was analyzed to identify potentially immunogenic peptides. The Phyre2 (Protein Homology/analogY Recognition Engine V 2.0; http://www.sbg.bio.ic.ac.uk/phyre2, accessed on 15 December 2021) and BepiPred-2.0 program (IEDB Analysis Resource) were used to analyze and predict the linear B-cell epitopes of the protein, and the similarity of candidate peptides to published ATAQ proteins was assessed. The identified immunogenic peptide was amplified from *HlATAQ* and inserted into a pCold-ProS2 plasmid (Takara Bio), after double digestion with *Xho*I and *Bam*HI. Upon confirmation of the plasmid insert by sequence analysis, *E. coli* BL21(DE3) was used for the transformation of pCold-ProS2-HlATAQ plasmid and protein expression. The non-inserted plasmid served as a control (rProS2).

Recombinant proteins were expressed according to the vector manufacturer’s protocols. In detail, transformed *E. coli* BL21(DE3) were cultured at 37 °C with vigorous shaking until the optical density at 600 nm reached 0.4, and cooled to 15 °C for 30 min. Afterward, induction for recombinant protein (rHlATAQ, rtHlATAQ, and rProS2) overexpression was initiated with 0.8 or 1 mM isopropyl β-D-1-thiogalactopyranoside, and culture at 15 °C for 24 h. Recombinant HlATAQ proteins were purified using a binding buffer supplemented with high salt concentration (20–40 mM Imidazole), Ni-NTA agarose (Qiagen), and Poly-Prep^®^ chromatography columns (Biorad, Hercules, CA, USA). rProS2 proteins were purified using Capturem ^TM^ His-Tagged Purification Maxiprep Columns (Takara Bio).

For the sequence confirmation prior to protein expression, constructed plasmids were first multiplied by transformation into the *E. coli* DH5 alpha strain, then purified and subjected to sequencing using the BigDye™ Terminator v3.1 Cycle Sequencing Kit (Applied Biosystems) and ABI Prism 3100 Genetic Analyzer (Applied Biosystems). Expression and purification of recombinant proteins were verified at each step, using sodium dodecyl sulfate-polyacrylamide gel electrophoresis (SDS-PAGE). SDS-PAGE results were visualized by the Coomassie blue staining method.

### 2.8. Vaccine Experiments

A small-scale vaccination trial was performed to evaluate the effectiveness of HlATAQ as an anti-tick vaccine. Two Japanese white rabbits, one immunized with recombinant HlATAQ and the other immunized with rProS2 protein (control rabbit) were experimentally infested with *H. longicornis*, and the conditions of ticks and rabbits were monitored. Rabbit immunization, tick infestation, and evaluation of vaccination effect were carried out as follows. Upon purification, the recombinant proteins were confirmed by a western blotting analysis using anti-ProS2 mouse monoclonal IgG as the primary antibody and horseradish peroxidase-conjugated sheep anti-mouse IgG as a secondary antibody. Afterward, the proteins were dialyzed by Slide-A-Lyzer™ Dialysis Cassettes (20K MWCO) (Thermo Fisher Scientific), and protein concentrations were measured by a fluorometer, Qubit 3.0 (Thermo Fisher Scientific), prior to immunization. 

For immunization, each rabbit received an intradermal injection of 300 µg recombinant HlATAQ or rProS2 protein adjuvanted with TiterMax gold (Sigma-Aldrich) in a 1:1 ratio. On day 14 after the primary immunization, each rabbit was boosted with the same dose of the antigen emulsified with the adjuvant. Prior to immunization, after each vaccination and 14 days after the tick challenge, blood was collected from the ear vein of the rabbits. Sera were prepared and their reactivity against recombinant HlATAQ or rProS2 antigens was verified by SDS-PAGE. In addition, sera antibody titers were measured using an enzyme-linked immunosorbent assay (ELISA) with sheep anti-rabbit IgG (H+L) as a secondary antibody.

Tick infestation was performed 16 days after the boost immunization. Each rabbit was infested with 30 female *H. longicornis* using the ear bag method [30]. Ticks were left to feed until engorgement. After dropping, each tick was weighed and then monitored for survival rate, egg laying, and subsequent hatching to larvae. To evaluate the effect of vaccination, the distribution of the blood-feeding period (days), body weight at engorgement (mg), pre-oviposition period (days), egg mass (mg), egg mass to body weight ratio, and egg hatching period (days) of ticks infested on recombinant HlATAQ-immunized rabbit were compared to the values obtained for the ticks infested on the control rabbit.

### 2.9. Statistical Analysis

The data obtained in the gene expression analysis, RNAi experiments, and vaccine experiments were analyzed using Microsoft Excel 2010, the statistical software R version 4.1.1 (R Foundation for Statistical Computing, Vienna, Austria), and SPSS Statistics for Windows, v26.0 (IBM Crop., Armonk, NY, USA). Differences between the two groups were analyzed using the Student’s *t*-test, Welch’s *t*-test, or the Mann–Whitney test according to the distribution of data. Comparisons of values obtained in more than two groups were performed using the one-way ANOVA test with a post-hoc Bonferroni for multiple pairwise comparisons or the Tukey Honestly Significant Difference test (Tukey’s HSD). A *p*-value of less than 0.05 was considered statistically significant.

## 3. Results

### 3.1. Identification of HlATAQ

Searching the *H. longicornis* midgut EST database, the ESTs of two cDNA clones (S02086B-17_A10 and S02086B-06_k17) with sequences sharing similarity with *R. microplus* Bm86 were identified. The EST of S02086B-17_A10 shared 86.36% identity with the Bm86 homologue previously characterized in *H. longicornis* (*Hl 86)* [23] and therefore was excluded from further analysis. The closest BLASTX match (64.56%) for the EST of S02086B-06_k17 was the ATAQ protein characterized in *Dermacentor reticulatus* ticks. Identities to the other ATAQ proteins ranged from 63.64% to 43.45%. S02086B-06_k17 was considered as the clone containing the BmATAQ homologue of *H. longicornis* and was sequenced to obtain the full-length cDNA sequence.

The complete nucleotide sequence of the cDNA was 2371 bp including a cap, a 5′ UTR, a protein coding sequence (ORF), a 3′UTR, and a poly (A) tail. The ORF was 1965 bp-long and ranged from an ATG start codon at position 118 to a TAG stop codon at position 2082 (Figure 1). The nucleotide and encoded protein were designated as *HlATAQ* and HlATAQ, respectively. When the *H. longicornis* genome was searched using the Genome Blast feature of the NCBI, *HlATAQ* showed 96.79% identity (95% coverage) with two sequences (CM023449, JABSTR010000004) registered as *H. longicornis* isolate HaeL-2018 chromosome 2 whole genome shotgun sequences. It is therefore plausible that *HlATAQ* locates on chromosome 2 of *H. longicornis*.

### 3.2. Characterization of HlATAQ Amino Acid Sequence

The protein encoded by *HlATAQ* was composed of 654 amino acids, had a predicted molecular weight of 70.6 kDa, and an isoelectric point of 4.6. The bioinformatic analysis showed that HlATAQ had a signal peptide at the N-terminus, a transmembrane domain, and an intracellular domain at the C-terminus. The YFNATAQRCYH signature peptide of the ATAQ protein family was found at the amino acid position 60-70, and seven EGF-like domains were identified. Five of the EGF-like domains completely fell into the pattern of the EGF-like region Cys-Xaa (3, 9)-Cys-Xaa (3, 6)-Cys-Xaa (8, 11)-Cys-Xaa (0, 1)-Cys-Xaa (5, 15)-Cys where Xaa is any amino acid other than cysteine and six cysteines are present in one EGF-like domain [16]. One EGF-like domain had six cysteines, however, instead of the expected three to nine amino acids; only one amino acid separated the first two cysteines. The remaining EGF-like domain contained four cysteines, which might correspond to a partial EGF-like domain as reported in other Bm86 and BmATAQ homologous proteins [16,23,31]. Eight potential N-glycosylation sites and 12 potential O-glycosylation sites were also identified in the HlATAQ sequence. The components and the map of HlATAQ are presented in Figure 2A,B.

Comparisons of the amino acid sequences of HlATAQ and ATAQ proteins from other tick species showed that all have six full and one partial EGF-like domains. HlATAQ had the highest molecular weight and number of glycosylation sites. Interestingly, except for *H. elliptica* ATAQ (HeATAQ) and *A. variegatum* ATAQ (AvATAQ) which had a GPI anchor, all ATAQ proteins had a transmembrane domain (Table 2). A comparison of the protein structure of HlATAQ with Bm86 and Hl86 showed that both HlATAQ and Hl86 had seven EGF-like domains while Bm86 had nine. HlATAQ had a similar molecular weight, but more glycosylation sites than Bm86 (Table 2).

### 3.3. BLAST and Phylogenetic Analyses of HlATAQ Amino Acid Sequence

The BLASTp analysis of HlATAQ did not return significant hits other than ATAQ and Bm86 homologous sequences. However, HlATAQ showed a low degree (<50%) of homology to previously known ATAQ proteins. The closest BLASTp match (47.79% identity; 99% coverage) of HlATAQ was the *D. reticulatus* ATAQ protein. The analysis of the proximity among tick genera showed that *H. longicornis* ATAQ shared higher identity/similarity with *Dermacentor* spp. and *H. m. marginatum* than other species including *H. elliptica* (Table 3).

In agreement, an amino acid sequence-based phylogenetic tree located HlATAQ on a divergent branch between the *Dermacentor* spp. clade and the *H. elliptica*/*A. variegatum* clade (Figure 3A). Conversely, in the tree inferred from the genes, *HlATAQ* formed a clade with *H. elliptica* and *A. variegatum* sequences (Figure 3B). The ATAQ sequences from the *Rhipicephalus* species, however, were consistently located in one clade, genetically distant from the *H. longicornis* sequence. The alignment of the deduced amino acid sequences of the ATAQ and Bm86 homologue sequences included in the phylogenetic analysis is shown in the Appendix A.

### 3.4. Expression Patterns of HlATAQ in Developmental Stages, Feeding Phases, and Tissues

The transcription levels of *HlATAQ* mRNA were measured in unfed and feeding or engorged specimens of all development stages. In addition, the variation of expression throughout feeding phases was examined in the midgut and Malpighian tissues which are the reported main expression sites of this protein [16].

*HlATAQ* mRNA was expressed constantly throughout the developmental stages of *H. longicornis* (Figure 4A). The highest expression level was recorded in engorged larvae, while the lowest was found in eggs. Notably, in larvae, nymphs as well as female ticks, blood-feeding significantly upregulated the expression of *HlATAQ*. However, the metamorphosis of engorged larvae to nymphs was followed by a significant decrease in *HlATAQ* transcripts. The analysis of each feeding phase in female ticks showed that *HlATAQ* expression steadily increased (*p* < 0.001) during the course of feeding, reached a peak, then decreased slightly with engorgement.

In the tick tissue analysis, *HlATAQ mRNA* transcripts were detected in both midgut and Malpighian tubules throughout the blood-feeding stages (Figure 4B). Expression levels in both tissues significantly peaked from the unfed period to the slow feeding period, and then decreased at full engorgement (*p* < 0.001).

### 3.5. Phenotype of HlATAQ Gene-Knockdown Ticks

The real-time PCR analysis of dsRNA-injected ticks showed that the *HlATAQ* mRNA expression in ticks injected with *HlATAQ*-2 dsRNA and *HlATAQ*-4 dsRNA were, respectively, 66.5% (*p* < 0.05) and 95.6% (*p* < 0.001) lower than those of *Luc* dsRNA-injected ticks (Figure 5A,B).

However, the suppression of *HlATAQ* gene expression by RNAi did not affect the tick phenotype. The recorded lengths of the blood-feeding period, pre-oviposition period, and oviposition to egg hatching period, the body weight at engorgement, and egg mass did not show any statistically significant differences between the RNAi and control ticks (Table 4).

### 3.6. Detection of HlATAQ Protein in Tick Tissues

Immunostaining with the anti-HlATAQ peptide serum was used to localize HlATAQ in the midgut and Malpighian tubules of female ticks. The tissues of the 5-day-fed ticks showed clearly higher fluorescence with the anti-HlATAQ serum than with the naïve mouse serum (Figure 6). Strong reactions to anti-HlATAQ serum were found throughout the cells of the midgut (Figure 6A) and Malpighian tubules (Figure 6B). In the Malpighian tubules, HlATAQ fluorescence intensity was particularly strong in the basal membrane (Figure 6B).

### 3.7. Expression, Purification, and Verification of Recombinant Proteins

The Phyre2 prediction software revealed that the folded structure of ATAQ is highly similar (78% coverage with 99% confidence) to apolipoprotein E receptor 2 (apoER2), a low-density lipoprotein. The N-terminal region of apoER2 reportedly contains ligand-binding domains [32,33] and therefore, we hypothesized that the N-terminal region of HlATAQ would be critical for the function of the protein. Further analysis of the seven EGF-like domains located in the N-terminal region of the HlATAQ protein, with the BepiPred-2.0 program, showed the presence of many B-cell epitopes throughout the domains. Therefore, along with the whole ORF of the protein, the N-terminal region containing EGF-like domains one–five (amino acid 29–249; length: 221 amino acids) was selected for the production of recombinant HlATAQ protein. The truncated protein shared 47–75% similarity with previously published ATAQ proteins and was PCR amplified from *HlATAQ* cDNA with the primer pairs rtHlATAQF and rtHlATAQR (Table 1).

Both the rHlATAQ and the rtHlATAQ plasmids were successfully transformed in *E. coli* BL21 (DE3). However, the expression of rHlATAQ was not successful and therefore only rtHlATAQ was expressed. After the removal of His and ProS2 tags, the aggregation of rtHlATAQ protein was high. Hence, to keep its water solubility, rtHlATAQ protein without removal of His-ProS2 tags was purified along with recombinant His-ProS2 protein as a control. The expected molecular weight of purified rtHlATAQ and rPros2 proteins were 48.3 kDa and 26.0 kDa, respectively (Appendix A). Purified rtHlATAQ and rPros2 proteins with the expected molecular weights were detected by anti-rPros2-mouse monoclonal IgG (Appendix A). Afterward, the purified proteins were dialyzed (Appendix A) and used for rabbit immunization.

### 3.8. Immune Response of Rabbit to Vaccination with Recombinant Proteins

The immune responses generated by rtHlATAQ and rPros2 vaccinations were verified by the identification of purified rtHlATAQ and rProS2 proteins with expected sizes using rabbit sera collected 11 days after the boost vaccinations (Appendix A). Since rtHlATAQ was purified without the removal of the His-ProS2 tags, we confirmed the specificity of the rabbit anti-rtHlATAQ antibodies. The analysis of the reactivity against rtHlATAQ and rProS2 of sera collected, before immunization, 11 days after primary vaccination, 11 days after boost-vaccination, and 14 days after tick challenge, showed that rtHlATAQ-immunized rabbit had antibodies specific to the protein. The reactivity of rtHlATAQ-immunized rabbit sera against rtHlATAQ antigen was significantly greater (*p* < 0.01) than that of rProS2-rabbit sera against rtHlATAQ and rProS2 antigens (Figure 7). 

### 3.9. Effect of Vaccination on H. longicornis

Figure 8 shows the effects of rHlATAQ vaccination. Compared to ticks fed on rProS2-immunized rabbit (control), the ticks fed on rtHlATAQ-immunized rabbit showed a significantly longer blood feeding period (Mann-Whitney U-test: *u* = 169, *z* = 4.14703, *p* < 0.01) and higher body weight at engorgement (Mann-Whitney U-test: *u* = 234.5, *z* = 3.17865, *p* < 0.01). Similarly, the pre-oviposition periods of ticks detached from rtHlATAQ-vaccinated rabbit tended to be significantly longer (Mann-Whitney U-test: *u* = 379.5, *z* = 4.14703, *p* < 0.01). The eggs laid by ticks fed on rtHlATAQ-immunized rabbit had a higher egg mass (Tukey HSD test: *Q* = 3.8122, *p* < 0.01) and took a significantly longer time to hatch than those from the control group (Tukey HSD test: *Q* = 4.2538, *p* < 0.01). Egg mass to body weight ratio values were not significantly different between the two tick groups. The blood-feeding period, body weight at engorgement, pre-oviposition period, egg mass, egg mass to body weight ratio, and egg hatching period of the two tick groups are detailed in Appendix A.

## 4. Discussion

In this study, ATAQ, a tick protein structurally related to Bm86 and a vaccine candidate, was investigated in *H. longicornis*.

Currently, characteristics of ATAQ proteins are available for tick species from *Rhipicephalus* (5 species), *Hyalomma* (one species), *Dermacentor* (2 species), *Amblyomma* (one species), and *Haemaphysalis* (one species) genera [16]. Two proteins (XM 037660470 and XP037574698), predicted from the whole genome sequencing of *R. sanguineus* (LOC119393444) and *D. silvarum* (LOC119456952) and registered in GenBank as “glycoprotein antigen BM86-like”, share a high sequence identity with ATAQ proteins. The HlATAQ in this study is therefore the second identification in *Haemaphysalis* genus and the 13th tick species in which the protein is confirmed. *Haemaphysalis longicornis* ATAQ shared most of the structural features of previously reported ATAQ proteins and appeared to be the longest of all. Previous studies [16,20] showed that except for one of the two *R. appendiculatus* ATAQ (RaATAQ-2), all *Rhipicephalus* spp. ATAQ had the same length, whereas, variation in length was observed in other genera. A similar variation in protein length had also been observed in Bm86 homologues [16] and may relate to the interspecies and inter-genera genetic variations of the ATAQ and Bm86 protein family. It is one of the reported causes of variation in the efficacy of Bm86- and ATAQ-based vaccines [20,34]. Another interesting finding was that, instead of a GPI anchor similar to *H. elliptica* ATAQ, HlATAQ had a TM anchor similar to *Rhipicephalus*, *Hyalomma,* and *Dermacentor* spp. Likewise, among Bm86 homologues, some proteins have GPI whereas others have a TM anchor [16]. Other protein families such as the cadherin superfamily [35] and the carcinoembryonic antigen (CAE) gene family also have GPI- and TM-anchored members. The fact that mutations in the transmembrane domain of the CAE family resulted in a shift from transmembrane- to GPI-anchorage [36] suggests that the TM found in *H. longicornis* ATAQ could be the ancestral domain, from which the GPI anchor of *H. elliptica* ATAQ could have derived following mutation events. 

GPI anchors of glycoproteins are rich in sphingolipids and cholesterol. They reportedly do not have uniform physical properties, interact with transmembrane proteins, and are involved in the transport of anchored proteins to the apical surface of epithelial cells [37,38]. Although it is unclear how the functions of ATAQ proteins differ depending on the type of anchor, a previous study suggests that replacing the GPI anchor with a transmembrane domain does not affect the function of certain proteins. In addition, some GPI-anchored proteins occur naturally in both GPI-anchored and transmembrane isoforms, which do not show functional dissimilarities. Meanwhile, the function of some GPI-anchored proteins is abolished when the GPI-anchor is replaced by a transmembrane protein domain [39]. Further studies on the function of ATAQ and Bm86 proteins and a comparison of GPI- and TM-anchored protein efficiency will certainly help in understanding the importance of the anchor. 

The multiple EGF-like domain is a key feature of the Bm86 and ATAQ protein families. The consensus sequence of the full EGF-like domain of the Bm86 family was first defined as Cys-Xaa (4, 8)-Cys-Xaa(3, 6)-Cys-Xaa(8, 11)-Cys-Xaa(0, 1)-Cys-Xaa(5, 15)-Cys based on the sequence of Bm86 from *R. microplus* [31]. Following the characterization of Bm86 and ATAQ proteins from several other tick species, the number of amino acids separating the first two Cys was updated to Xaa (3, 9) [16]. Based on the EGF-like domains of HlATAQ, we suggest the number of amino acids separating the first two Cys is updated to Xaa (1, 9). Noteworthy, improvement of the consensus formula always occurred in the same section of the protein. Whether this variation has an impact on the protein function remains unclear and could be an interesting topic for further investigation.

The finding that the *HlATAQ* gene is expressed at all developmental stages is in accordance with the data of previously identified ATAQ proteins [16]. However, the expression patterns were different. Compared to *BmATAQ,* the *HlATAQ* transcript level was more variable throughout the developmental stages. Meanwhile, opposite to *RaATAQ*, the expression level of *HlATAQ* was lower in unfed than feeding ticks [16]. These differences might indicate tick species-based variations of the function of ATAQ. 

To our knowledge, this study is the first evaluation of ATAQ protein transcript levels in tissues at different feeding phases. The expression of the HlATAQ gene in tissues was performed only in the midgut and Malpighian tubule for two reasons: (1) previously identified ATAQ proteins were exclusively expressed in these tissues; (2) since the midgut and the Malpighian tubule are, respectively, the organ in charge of blood digestion and osmotic pressure regulation [40], ATAQ expression levels in these tissues could be useful to evaluate the potential effect of an ATAQ-based vaccine. The observed variation of expression level in the midgut and Malpighian tubule indicates a relationship between blood-feeding and HlATAQ expression kinetics. We, therefore, inferred that disturbing the HlATAQ expression would affect tick blood meal and the related life parameters.

Our immunostaining experiments provide clarification on the localization of ATAQ in the midgut and Malpighian tubules. Extracellular proteins with EGF-like domains are generally involved in blood coagulation and complement cascades or are associated with the regulation of cell growth [31]. Bm86 reportedly resembles the proteins involved in cell growth and is expressed by stem cells and/or prodigest cells of the midgut epithelium [41,42]. ATAQ and Bm86 structural similarities may explain the localization of HlATAQ throughout the midgut cell. Meanwhile, because Malpighian tubules are derived from the endoderm and are thought to have originated as an extension of the intestine [43], their histological similarity with the midgut may explain the observed localization of HlATAQ. The high fluorescence intensity in Malpighian tubules’ basal membrane supports the expression of ATAQ protein by stem cells as hypothesized by Nijhof et al. [16]. Bm86 proteins from Ixodids were reportedly exclusively expressed in the midgut, but the argasid tick Bm86 homologue (Os86) could be identified in both midgut and Malpighian tubules. Although currently characterized ATAQ proteins were found in both midgut and Malpighian tubules, studies on other tick tissues and ATAQ proteins from other tick species are required for a thorough understanding of the distribution of these proteins.

The structure and expression patterns of ATAQ protein support a role in blood feeding. Therefore, the fact that *HlATAQ*-silenced ticks did not phenotypically differ from control ticks was surprising. It was, however, expected because, in the previous study, silencing of *R. evertsi evertsi* ATAQ did not result in a significantly different phenotype [16]. Similarly, no significant effect was obtained in RNAi of Bm86 [44], Ree86 alone, or Ree86 and ReeATAQ [16]. In contrast, silencing of Hl86 resulted in ticks characterized by significantly reduced weight at engorgement, but a similar blood feeding period, egg weight, or egg hatching ability with the control ticks [23]. Though it is unclear why the effects of silencing Hl86 and HlATAQ differed, these results may indicate that Bm86/ATAQ are components of a group of proteins with similar functions for which expression can be compensated in case of downregulation as hypothesized for some salivary gland genes [45].

The results of rabbit immunization with rtHlATAQ protein (domains 1–5), in agreement with the protective effect reported for the synthetic *Rhipicephalus* ATAQ peptide [19], support the potential of the ATAQ protein as an anti-tick vaccine. The physiological functions of ATAQ EGF-like domains are not well understood. However, it has been reported that human EGF-like domains have different functional roles and that derived forms exist [46]. ApoER2, for example, is known for its role in platelet aggregation and hemostasis, with its EGF domains serving as ligand-binding domains [47,48]. Meanwhile, Bm86 has been described as a cell membrane-bound ligand potentially transmitting positional or cell-type information to adjacent cells and influencing the cell lineage of those adjacent cells [31]. ATAQ proteins might have similar functions. Hence, in this study, we targeted the EGF-like domains, the potential ligand-binding region of HlATAQ, with the hope that antibodies directed against it would affect protein functions, leading to a vaccine effect. Immunological experiments using the recombinant EGF-like domain of the Bm86 ortholog of *Ixodes scapularis* (Is86), showed an effect on *I. scapularis* blood sucking and molting [49]. It is therefore reasonable to presume that antibodies targeting the EGF-like domain of HlATAQ could have generated the physiological effect observed on the ticks. Our hypothesis is that like the Bm86 vaccine destroys the midgut epithelium [31], anti-ATAQ antibodies may reach the midgut from the host blood and not only destroy the midgut but also damage the Malpighian tubule via the hemolymph. Such actions would have disturbed the blood-feeding process resulting in a longer feeding time to compensate for the defect, with consecutive effects on the weight at engorgement, delays in egg laying, higher egg mass, and delayed egg hatching. Noteworthy, contrasting with the effect of recombinant ATAQ, vaccination with recombinant Bm86 led to a reduction in the number of engorged ticks, lower engorgement weights, and a decrease in the number of oviposited eggs [16]. 

The results reported here give a hint at the immunogenic potential of HlATAQ, but are not fully conclusive. The experiments were performed on only one rabbit, and there were no repeats. Previous tick vaccine studies indicated protein genetic diversity [20,50] and the balance between host complement system-IgG antibodies’ collaborative action and tick protease–protease inhibitors [51] as factors determining the protective effect of Bm86/ATAQ-based vaccines. Thus, more elaborated and comprehensive studies covering HlATAQ genetic diversity, investigating several vaccination designs (single or several recombinant proteins, type of adjuvant, number of boost vaccinations), and the larger number of vaccinated animals are needed for the clarification of ATAQ potential as a commercial vaccine.

## 5. Conclusions

To sum up, the full-length cDNA sequence of the ATAQ gene of *H. longicornis* was identified using the midgut cDNA library of semi-engorged female ticks. The corresponding protein was localized, functionally characterized, and then evaluated in a vaccine experiment. HlATAQ, which shared the same structure with previously identified ATAQ proteins, was located in the midgut digestive cells and Malpighian tubule cells and expressed at all life stages. Its expression seemed upregulated by blood-feeding; however, gene silencing did not affect tick phenotype. Ticks fed on a rabbit immunized with recombinant HlATAQ tended to have an extension of the blood feeding period, accompanied by changes in pre-oviposition and egg hatching periods. Based on these findings, ATAQ is considered to be an anti-tick vaccine candidate which deserves future studies to discover its role in tick feeding and potential use for tick control.

## Figures and Tables

**Figure 1 microorganisms-11-00822-f001:**
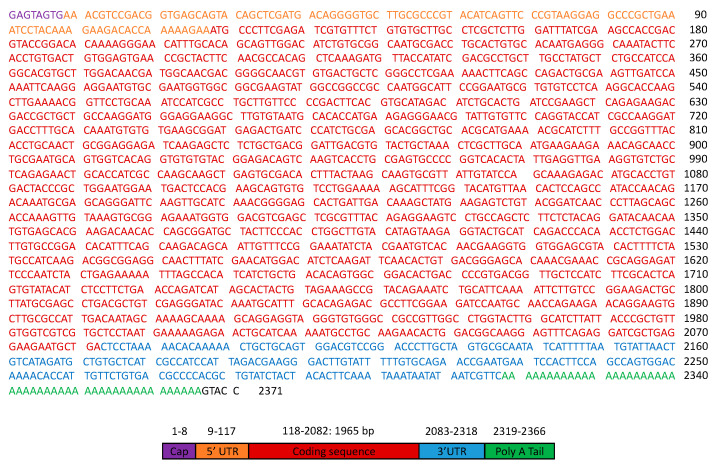
Nucleotide sequence of HlATAQ cDNA full length (2371 bp). The components of the cDNA were identified by referring to the features of cDNA produced using the vector capping method [22] and examining the amino acid translation. UTR: untranslated region.

**Figure 2 microorganisms-11-00822-f002:**
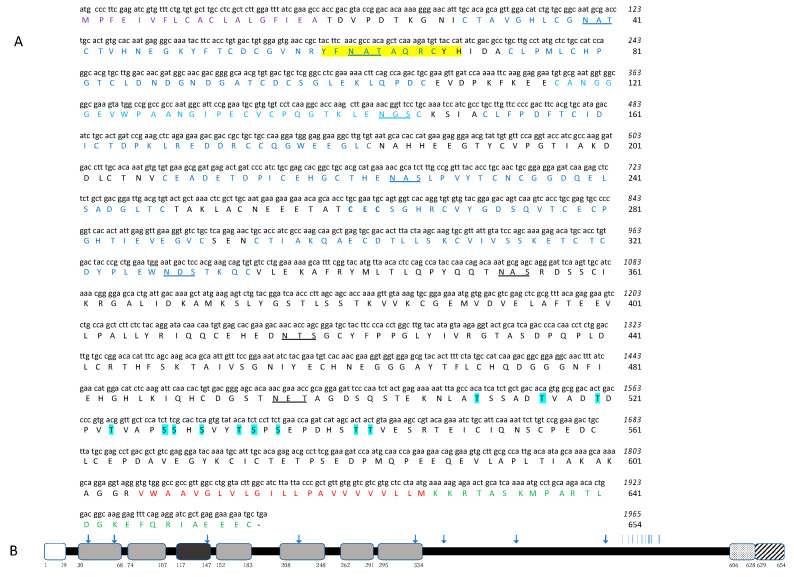
Structure of HlATAQ protein. (**A**) Nucleotide sequence (lowercase letters) and derived amino acid sequence (capital letters) of HlATAQ. The putative signal peptide (purple), epidermal growth factor (EGF)-like domains (deep blue), partial EGF-like domains (light blue), a signature peptide of ATAQ protein family (yellow highlight), intracellular domain (green), and transmembrane domain (red) are indicated. The discrepancy with the previously known pattern of the EGF-like domain is indicated in bold letters. The sites for the potential N-linked glycoprotein are underlined while the sites for the potential O-linked glycoprotein are indicated with blue highlights. (**B**) Schematic diagram of the amino acid sequence of HlATAQ. The signal peptide (white box), epidermal growth factor (EGF)-like domains (light grey boxes), partial EGF-like domains (dark grey boxes), transmembrane (TM) domain (dotted box), and intracellular domains (striped box) are indicated. Potential O-linked glycosylation sites are indicated by a vertical line and the potential N-linked glycosylation by arrows. The numbers correspond to the amino acid (AA) positions of the start and end of each protein domain.

**Figure 3 microorganisms-11-00822-f003:**
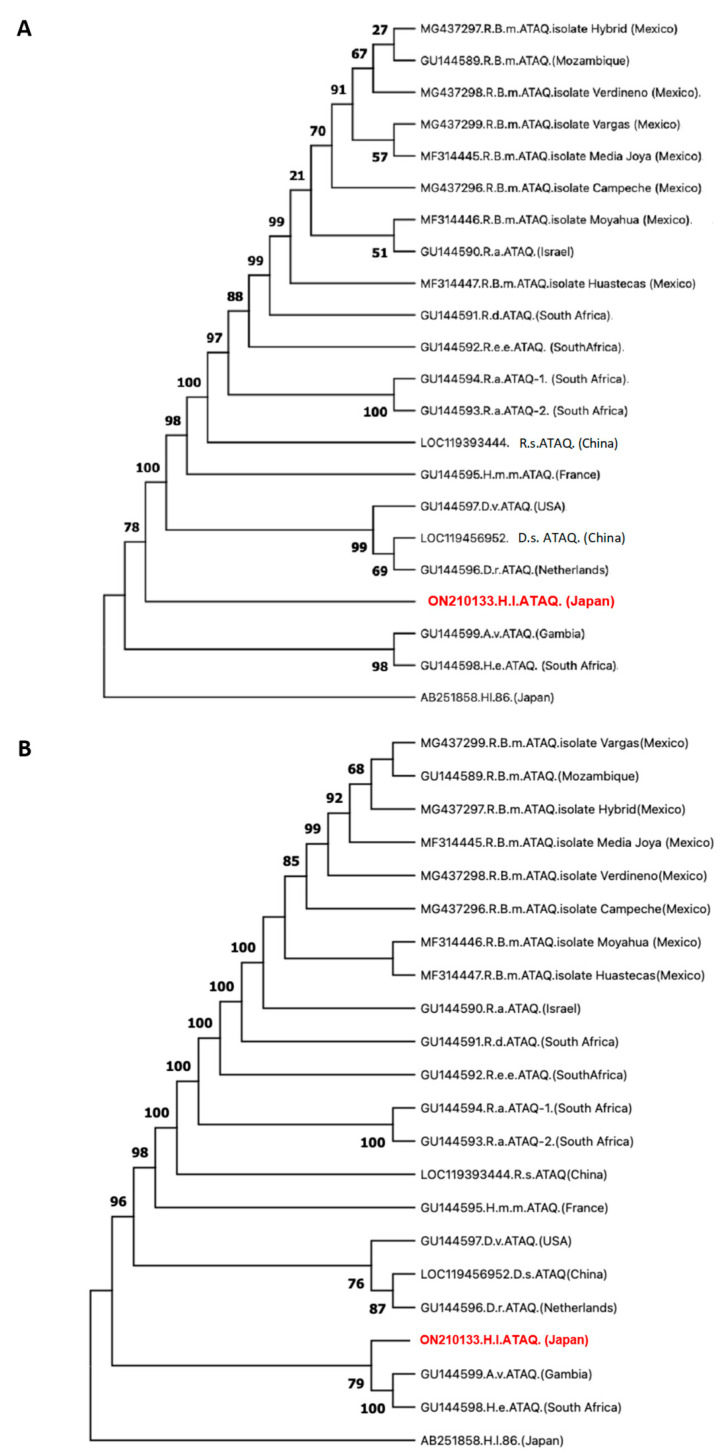
Phylogenetic relationship of ATAQ sequences. (**A**) Phylogenetic tree showing the relationship between HlATAQ and ATAQ sequences from other tick genera as estimated by maximum likelihood analysis. This cladogram covers the full length of 22 protein sequences and was constructed in MEGA X using the Jones-Taylor-Thornton model and a discrete Gamma distribution with invariable variation to model evolutionary rate differences among sites (+G+I). The Bm86 homologue from *Haemaphysalis longicornis* (Hl86) was used as an outgroup. Bootstrap values are shown as percentages at nodes based on 1000 replicates. The GenBank accession number and country of origin of each strain are indicated. The HlATAQ sequence generated in the current study is shown in red. (**B**) Phylogenetic tree showing the relationship between *HlATAQ* and *ATAQ* sequences from other tick genera as estimated by maximum likelihood analysis. This cladogram covers the full length of 22 gene sequences and was constructed in MEGA X using the Kimura 2-parameter model and a discrete Gamma distribution to model evolutionary rate differences among sites (+G). The Bm86 homologue from *Haemaphysalis longicornis* (Hl86) was used as an outgroup. Bootstrap values are shown as percentages at nodes based on 1000 replicates, and values lower than 50% were omitted. The GenBank accession number and country of origin of each strain are indicated. The *HlATAQ* sequence generated in this study is indicated in red.

**Figure 4 microorganisms-11-00822-f004:**
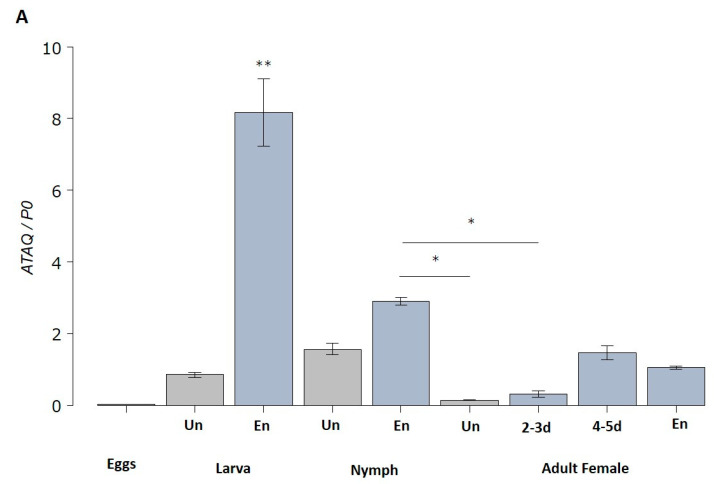
Quantitative reverse transcriptase-PCR analysis showing the transcription levels of *HlATAQ* mRNA in *H. longicornis* ticks. (**A**) Expression profiles of *HlATAQ* at different developmental stages and feeding phases. Total RNA was extracted from eggs, unfed larvae, unfed nymphs, unfed female, engorged larvae, engorged nymphs, 2–3 days-fed female ticks (2–3 d), 4–5 days-fed female ticks (4–5 d), and fully engorged female ticks. The cDNA levels were indexed to the *H. longicornis P0* gene. The data are presented as mean ± standard error. Un: unfed; En: engorged. * *p* < 0.05, ** *p* < 0.001. (**B**) Transcription levels of *HlATAQ* mRNA in the midgut and Malpighian tissues at different phases of blood feeding. Total RNA was extracted from the midgut and Malpighian tissues of unfed female ticks (Un), 2-days-fed female ticks (2 d), 4-days-fed female ticks (4 d), 6-days-fed female ticks (6 d), and engorged female ticks (En). The cDNA levels were indexed to the *H. longicornis P0* gene. The data are presented as mean ± standard error. * *p* < 0.05, ** *p* < 0.001.

**Figure 5 microorganisms-11-00822-f005:**
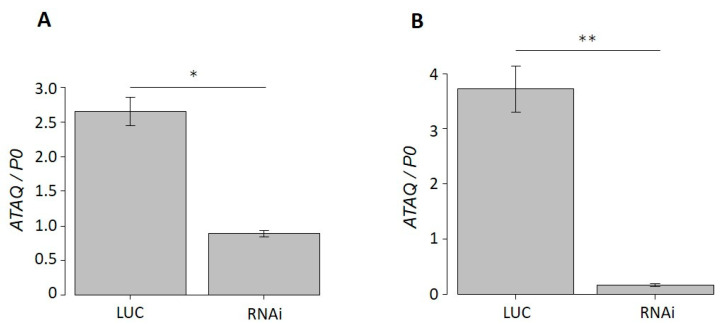
Efficiency of suppression of *HlATAQ* gene expression by RNA interference (RNAi) in *H. longicornis*. (**A**) *HlATAQ*-2 dsRNA-based gene suppression. (**B**) *HlATAQ*-4 dsRNA-based gene suppression. In each RNAi and control tick group, total RNA was purified from 4 day fed ticks. The gene silencing was assessed by real-time PCR. The *HlATAQ* expression was normalized to the *H. longicornis* P0 mRNA expression. Data represent the mean ± standard error. *HlATAQ* expressions in RNAi-treated ticks (RNAi) were significantly lower (A: 66.5%; B: 95.6%) than those of control ticks (LUC). Statistical analysis was performed by Welch’s test. * *p* < 0.05 ** *p* < 0.001.

**Figure 6 microorganisms-11-00822-f006:**
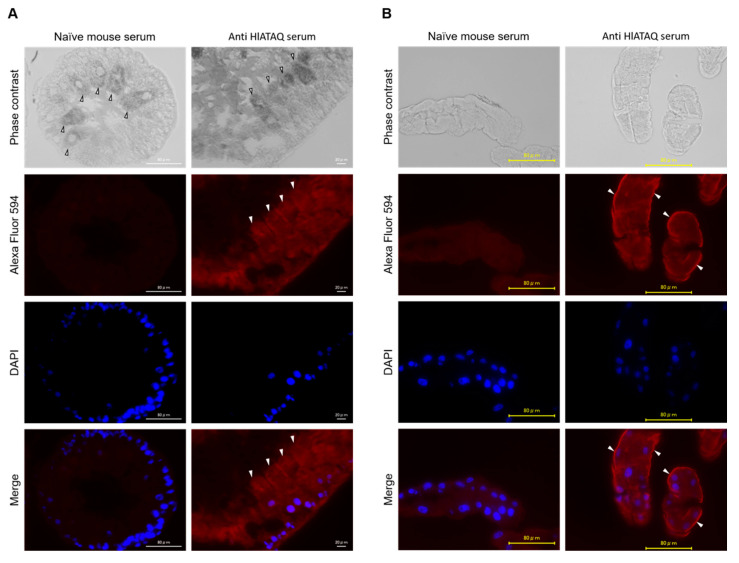
Localization of HlATAQ protein in the midgut (**A**) and Malpighian tubules (**B**) of *H. longicornis* with an anti-HlATAQ serum. The section at 400× magnification from 5-day fed female ticks. Immunostaining was performed with mouse anti-HlATAQ peptide antibodies (1:100 diluted) and with naïve mouse serum (1:100 diluted) which was used as a control. HlATAQ presence is displayed in red color by Alexa Fluor^®^ 594; the cell nuclei are displayed in blue by DAPI. The arrow indicates a midgut digestive cell (**A**) and the basal membrane of Malpighian tubules (**B**). Scale bars: 80 µm.

**Figure 7 microorganisms-11-00822-f007:**
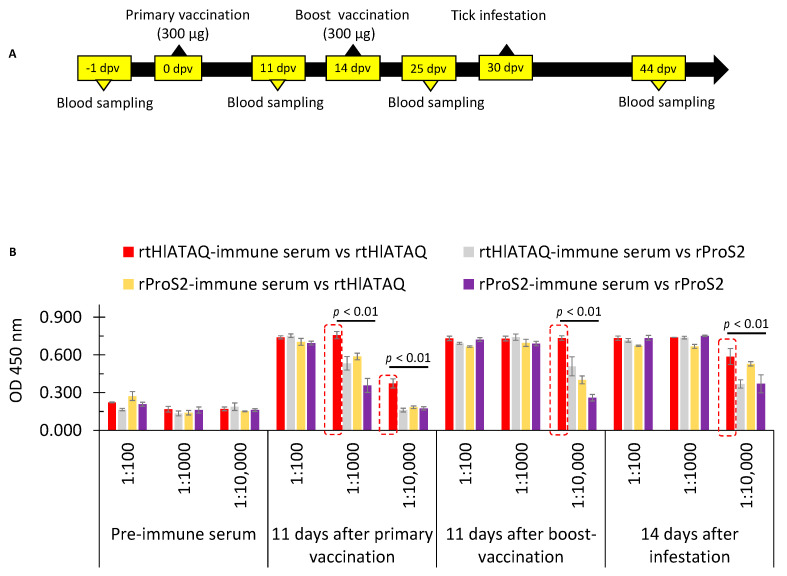
Evaluation of rabbit immune response after vaccination with rtHlATAQ and rProS2. (**A**) Schematic presentation of vaccination and blood samplings. (**B**) Antibody levels and reactivity of rtHlATAQ serum and rProS2 serum against rtHlATAQ and rProS2 proteins as estimated by ELISA. Antibody titers of pre-immune, 11 days after primary vaccination, 11 days after boost-vaccination, and 14 days after tick infestation are shown at various dilutions (1:100, 1:1000; 1:10,000). A comparison of OD values was performed using the one-way ANOVA test with a post-hoc Tukey Honestly Significant Difference Test. In each comparison, red rectangles indicate the highest OD values of reactivity against rtHlATAQ or rProS2 (20 ng each well) for sera diluted at the same ratio. A significant difference was considered when the *p*-value was less than 0.05.

**Figure 8 microorganisms-11-00822-f008:**
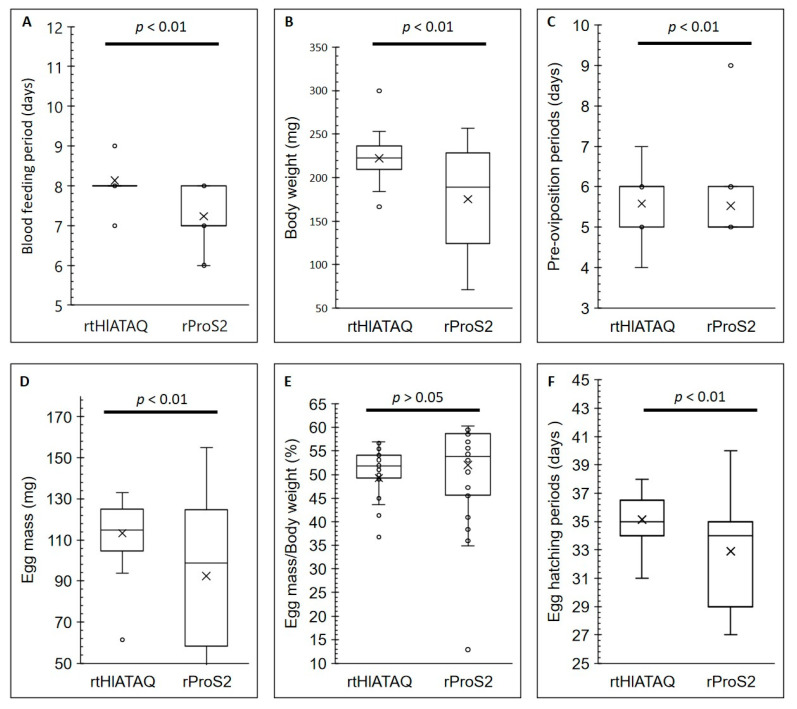
Effects of rHlATAQ vaccination on *H. longicornis* infestation in a rabbit. The results compare the parameters of rHlATAQ-immunized rabbit fed ticks and rProS2-immunized rabbit fed ticks (control). The results are shown as box plots. (**A**) Blood-feeding periods. (**B**) Body weight at engorgement. (**C**) Pre-oviposition periods. (**D**) Egg mass. (**E**) Egg mass to body weight ratio. (**F**) Egg hatching periods. A comparison between the two groups for each parameter was performed using the Mann–Whitney test or Tukey Honestly Significant Difference Test. A *p*-value less than 0.05 was considered as a significant difference.

**Table 1 microorganisms-11-00822-t001:** List of primers used in this study.

Experiment	Primer Name	Nucleotide Sequence (5′-3′)	At the Position of the Sequence ^a^
Sequencing			
	pGCAP 1–2	ACTGCTCCTCAGTGGATGTT	
	pGCAP-F	CTCAGTGGATGTTGCCTTTAC	
	M13 (-21)	TGTAAAACGACGGCCAGT	
	HlATAQ-1	GAGGAATGTGCGAATGGTGG	343–362
	HlATAQ-2	GAGGTTGAAGGTGTCTGCTCA	856–876
	HlATAQ-3	ACAAACGAAACCGCAGGAGA	1483–1502
	T3 primer	ATTAACCCTCACTAAAGGGA	
	pGCAP-R	GCATTCTAGTTGTGGTTTGTCC	
Real-time PCR			
	HlP0 F	CTCCATTGTCAACGGTCTCA	
	HlP0 R	TCAGCCTCCTTGAAGGTGAT	
	HlqPCR ATAQ-1 F	TCGGAAGATCCAATGCAACCAG	1738–1759
	HlqPCR ATAQ-1 R	TTAGGAGCACGACGACCACAAC	1861–1882
dsRNA synthesis			
HlATAQ-4 dsRNA	T7-HlATAQ-4 F	**TAATACGACTCACTATAGG**TGTGACTGTGGAGTGAACCG	157–176
	T7-HlATAQ-4R	**TAATACGACTCACTATAGG**GAGCTCTTGATCTCCTCCGC	704–723
HlATAQ-2 dsRNA	T7-HlATAQ-2 F	**TAATACGACTCACTATAGG**GAGGTTGAAGGTGTCTGCTCA	856–876
	T7-HlATAQ-2 R	**TAATACGACTCACTATAGG**CGCCGTCTTGATGGCATAGA	1410–1429
Recombinant protein expression			
rHlATAQ	rHlATAQF	CGCGGATCCATGCCCTTCGAGATCGTG	
	rHlATAQR	GGTGCTCGAGTCAGCATTCTTCCTCAGCG	
	T7 promoter	TAATACGACTCACTATAGGG	
	rHlATAQ middle	CGCCAAGGATGACCTTTGCAC	
	T7 terminator	ATGCTAGTTATTGCTCAGCGG	
rtHlATAQ	rtHlATAQF	ACCCTCGAGATTTGCACAGCAGTTGGAC	
	rtHlATAQR	TTCGGATCCTCAAGTACACGTCAATCCGTCAGC	

^a^ Position of the primers on HlATAQ nucleotide sequence. The T7 promoters are in bold letters. rHlATAQ: recombinant protein covering the whole ORF of HlATAQ. rtHlATAQ: recombinant protein covering a portion of the whole ORF of HlATAQ.

**Table 2 microorganisms-11-00822-t002:** Comparison of HlATAQ protein structure with other ATAQ proteins, Bm86 and Hl86.

Protein	Tick Species	Accession No.	AA No.	MW	pI	Glycosylation (N-Linked/O-Linked)	EGF Domains (Full/ Partial)	Anchor
HlATAQ	*H. longicornis*	ON210133	654	70.6	4.6	8/12	6/1	TM
BmATAQ	*R.* (*B.*) *microplus*	GU144589	605	66.6	4.82	8/2	6/1	TM
BdATAQ	*R.* (*B.*) *decoloratus*	GU144591	605	66.5	5.16	8/4	6/1	TM
BaATAQ	*R.* (*B.*) *annulatus*	GU144590	605	66.4	5.05	8/2	6/1	TM
ReeATAQ	*R. evertsi evertsi*	GU144592	605	66.4	4.95	8/3	6/1	TM
RaATAQ-1	*R. appendiculatus*	GU144594	605	66.7	5.42	7/3	6/1	TM
RaATAQ-2	*R. appendiculatus*	GU144593	561	61.6	5.33	8/1	6/1	TM
HmATAQ	*Hy. m. marginatum*	GU144595	601	65.5	5.18	5/1	6/1	TM
DrATAQ	*D. reticulatus*	GU144596	596	64.7	4.79	9/2	6/1	TM
DvATAQ	*D. variabilis*	GU144597	598	65.0	4.84	6/4	6/1	TM
HeATAQ	*H. elliptica*	GU144598	597	65.6	5.47	3/17	6/1	GPI
AvATAQ	*A. variegatum*	GU144599	522	57.5	5.04	4/1	6/1	GPI
Hl86	*H. longicornis*	AB251858	594	66.6	5.6	5/10	6/1	GPI
Bm86	*R.* (*B.*) *microplus*	M29321	650	71.72	5.6	4/2	8/1	GPI

Accession No., GenBank accession number; AA No., number of amino acids in the protein; MW, molecular weight in Kilodalton; pI, isoelectric point; TM, transmembrane domain; GPI, glycosyl-phosphatidylinositol; R., *Rhipicephalus*, B., *Boophilus*; Hy. m., *Hyalomma marginatum*; D., *Dermacentor*; H., *Haemaphysalis*; A., *Amblyomma*. This table was constructed by adding the information on HlATAQ (this study) to the data from Nijhof et al. [16] and Liao et al. [23].

**Table 3 microorganisms-11-00822-t003:** Identity/similarity between the full amino acid sequences of HlATAQ and other species ATAQ proteins deposited in the GenBank.

ATAQ Sequences		Homology to HlATAQ (%)	
Tick species	Country	Identity	Similarity
*D. reticulatus*	Netherlands	48.61	53.43
*D. variabilis*	USA	47.59	51.38
*Hy. m. marginatum*	France	46.86	51.82
*R.* (*B.*) *decoloratus*	South Africa	44.23	49.34
*R. sanguineus*	China	44.08	49.63
*R.* (*B.*) *microplus*	Mozambique	43.94	49.34
*R.* (*B.*) *microplus*	Mexico (Media Joya)	43.94	49.19
*R.* (*B.*) *microplus*	Mexico (Moyahua)	43.94	49.34
*R.* (*B.*) *microplus*	Mexico (Campeche)	43.94	49.19
*R.* (*B.*) *microplus*	Mexico (Hybrid)	43.94	49.34
*R.* (*B.*) *microplus*	Mexico (Verdineno)	43.94	49.34
*R. evertsi evertsi*	South Africa	43.79	49.05
*R.* (*B.*) *microplus*	Mexico (Vargas)	43.64	49.05
*R.* (*B.*) *microplus*	Mexico (Huastecas)	43.50	49.05
*R.* (*B.*) *annulatus*	Israel	43.35	48.46
*R. appendiculatus*	South Africa	43.35	49.63
*R. appendiculatus*	South Africa	40.72	46.27
*D. silvarum*	China	40.72	42.91
*A. variegatum*	Gambia	34.16	38.68
*H. elliptica*	South Africa	32.70	37.51

R., *Rhipicephalus*; B., *Boophilus*; Hy. m., *Hyalomma marginatum*; D., *Dermacentor*; H., *Haemaphysalis*; A., *Amblyomma*. The *R. sanguineus* and *D. silvarum* sequences used in this analysis are predicted protein sequences (XM 037660470 and XP037574698, respectively). They were derived from genomic sequences (LOC119393444 and LOC119456952, respectively) by automated computational analysis and are recorded in the GenBank as “glycoprotein antigen BM86-like”.

**Table 4 microorganisms-11-00822-t004:** Effects of *HlATAQ* knockdown by RNAi on adult tick engorgement and oviposition.

Groups	No. of Ticks	Blood Feeding Period (Days)	Body Weight of Engorged Ticks (mg)	Egg Mass/Body Weight (%)	Pre- Oviposition Period (Days)	Egg Hatching Period (Days)
Control-1	22	7.2 ± 0.2	183.4 ± 8.9	41.2 ± 1.8	6.1 ± 0.2	38.9 ± 0.5
RNAi/*HlATAQ-2* dsRNA	19	6.9 ± 0.2	182.8 ± 10.2	45.0±2.0	5.9 ± 0.2	39.4 ± 0.5
Control-2	22	7.2 ± 0.2	231.8 ± 8.8	56.1 ± 0.8	5.9 ± 0.2	36.4 ± 0.3
RNAi/*HlATAQ-4* dsRNA	22	7.5 ± 0.2	232.5 ± 7.5	54.7 ± 0.6	5.8 ± 0.2	36.5 ± 0.4

The data are shown as mean of +/− standard errors. RNAi/*HlATAQ*-2 dsRNA, *HlATAQ*-2 double-stranded RNA (dsRNA)-injected *H. longicornis* females; RNAi/*HlATAQ*-4 dsRNA, *HlATAQ*-4 dsRNA-injected *H. longicornis* females; Control-1 and Control-2, *H. longicornis* females injected with dsRNA of the firefly luciferase gene.

## Data Availability

All of the relevant data are provided in the form of regular figures, tables, and Appendix A.

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
