# Peer review of "Identification and Characterization of *Rhipicephalus microplus* ATAQ Homolog from *Haemaphysalis longicornis* Ticks and Its Immunogenic Potential as an Anti-Tick Vaccine Candidate Molecule"

_microorganisms, 2023, doi:10.3390/microorganisms11040822_

Round 1

Reviewer 1 Report

The Adjou Moumouni and collaborators' manuscript is clear, understandable, and well-written. The study has required a significant amount of work, the identification and characterization of an ATAQ gene homolog and its product protein from the tick Haemaphysalis longicornis was performed, and also the protective effect of this protein was also tested. While the overall findings are compelling, there are a few shortcomings that should be easily addressed before publication.

Mayor considerations:

  1. The authors identified and isolated a gene from Haemaphysalis longicornis related to the ATAQ family that are a potential candidate as a protective antigen against ticks. The complete mRNA was isolated and sequenced in figure 1 authors showed the features of the HIATAQ  transcript including the cap, the 5´UTR, the 3´UTR, and the poly (A) tail. The authors must clarify which bioinformatic tools were used to identify the ORF and the other sequences described in the figure. As the Haemaphysalis longicornis genome has been published authors could analyze the complete gene structure and the chromosome localization to offer a full picture of this gene.
  2. The authors also make an in silico characterization of the HIATAQ protein predicting the signal peptide and EGF-like domains in the material and method section authors described that they used a domain search using PRATT, this program find conserved patterns in the amino acid sequences, do this conserved patterns that the authors described as EGF-like domains were also validated using InterPro or other tools for domain identification? 
  3. As a supplementary figure authors could add an alignment of the ATAQ and Bm86 sequences included for the phylogenetic analysis, showing the conserved domains and motifs
  4. Table 2 is showing a comparison between protein characteristics, the name of column 1 should be changed to protein. The accession number of HIATAQ gene does not retrieve the sequence in GenBank.
  5. Regarding the expression patterns of the HlATAQ gene at different developmental stages and feeding phases, the authors described that the data were normalized using the HIPO gene as a reference gene. Data could be shown as relative expression as in Figure 5 or explained in more detail how the data was normalized.  
  6. The authors purified the recombinant rtHlATAQ in figure S1 but unlike the rProS2 protein, there are many more contaminating bands. Why didn't the authors make more chromatographic steps to gain more purity of their protein? It will be important to discuss this. 
  7. As the Authors obtained antibodies against the recombinant rtHlATAQ protein, why did they not analyze the protein levels in their RNA interference assays by western blot or immunofluorescent? This could be import to support the possible explanations for not finding any effect when silencing the gene

Minor considerations:

The manuscript is well written but there is an incomplete sentence:

  1. In the introduction in line 73, the sentence seems incomplete: “Current efforts to develop a new generation of anti-tick 71 vaccines focus on identifying candidate antigens and combinations of antigens that will 72 be more consistently effective than Bm86 has been……” 

Reviewer 2 Report

In this paper, the authors carry out a thorough description and analysis of an anti-tick vaccine candidate protein ATAQ, from an important tick for veterinary health, H. longicornis. This includes sequence analysis and comparison with ATAQ and Bm86 from other tick species, localisation analysis, expression profiles through tick life stages and feeding, RNAi knockdown experiments, and a small vaccine trial. This impressive body of work gives an indication of the role of this protein in tick biology. The introduction gives an excellent background to the research. Methods are described very thoroughly. Results are clearly presented, and the Discussion provides a good summary of the experiments and state of the field.

Despite the interest of this protein as a vaccine candidate, RNAi knockdown produced no phenotype and the small vaccine trial resulted in relatively minor changes to tick feeding and egg laying success, so I feel that the authors are over-enthusiastic about the potential of this protein to be developed as a vaccine. In the field, the observed effects on blood feeding and egg hatching would have very little effect on the tick life cycle, as the extensions to these were only a few days. However, as the authors concede, the vaccine trial included was performed on a single rabbit, so further work is needed to evaluate this protein as a vaccine candidate.

I would recommend that in the Conclusion (lines 734-735) that authors be more realistic in their discussion of the promise of ATAQ as an anti-tick vaccine candidate. It will certainly be interesting to discover the role this protein plays in tick feeding and this paper makes significant progress towards that goal.

Further minor edits are suggested below:

line 236: spelling "..fourth coxae..." and "...glass slide..."

line 240: correct to "The rabbits were monitored daily."

line 351 & 355: these should say "homologue" not "homologous"

line 380: should say "Eight potential...."

Figure 4: Since these graphs are showing expression relative to P0, the x axes should be labelled more descriptively, for example "relative expression of ATAQ" or "ATAQ / P0" 

Figure 5: Was there any difference between expression in ticks collected at 4 days fed and 20 days after oviposition? Do these graphs show the expression in combined tick RNA from the two timepoints?

line 542: could the unsuccessful expression of rHlATAQ be because it was expressed from a different expression vector?

line 616: should say "homologues" not "homologous"

References - make sure all the species names in references are italicised
